# Scalable total synthesis and comprehensive structure–activity relationship studies of the phytotoxin coronatine

Mairi M. Littleson[1], Christopher M. Baker[2], Anne J. Dalençon[2], Elizabeth C. Frye[2], Craig Jamieson[1], Alan R. Kennedy[1], Kenneth B. Ling[2], Matthew M. McLachlan[2], Mark G. Montgomery[2], Claire J. Russell[2] & Allan J.B. Watson[3]

Natural phytotoxins are valuable starting points for agrochemical design. Acting as a jasmonate agonist, coronatine represents an attractive herbicidal lead with novel mode of action, and has been an important synthetic target for agrochemical development. However, both restricted access to quantities of coronatine and a lack of a suitably scalable and flexible synthetic approach to its constituent natural product components, coronafacic and coronamic acids, has frustrated development of this target. Here, we report gram-scale production of coronafacic acid that allows a comprehensive structure–activity relationship study of this target. Biological assessment of a >120 member library combined with computational studies have revealed the key determinants of potency, rationalising hypotheses held for decades, and allowing future rational design of new herbicidal leads based on this template.

[1] Department of Pure and Applied Chemistry, University of Strathclyde, 295 Cathedral Street, Glasgow G1 1XL, UK. [2] Syngenta, Jealott's Hill International Research Centre, Bracknell, Berkshire RG42 6EY, UK. [3] EaStCHEM, School of Chemistry, University of St Andrews, North Haugh, St Andrews KY16 9ST, UK. Correspondence and requests for materials should be addressed to A.J.B.W. (email: aw260@st-andrews.ac.uk)

Food security is recognised as a global concern due to a growing population increasing food consumption and various factors that diminish production, such as arable land desertification and infestation by pests[1]. The requirement for effective herbicides for improved weed control and crop yield is essential to meet global food demand[2]. Resistance to traditionally used herbicides is an increasing problem and there is increasing regulatory pressure on the current crop protection products available to the farmer[3], which has resulted in a pressing need for the development of novel and safe agrochemicals with new modes of action [4]. In this regard, natural products are valuable starting points for agrochemical design as they often allow the targeting of distinct biological space;[5,6] mesotrione is a pertinent example of how natural assets can be leveraged to new herbicidal agents[7].

Coronatine (COR, 1; Fig. 1a) is produced by several strains of *Pseudomonas syringae* and has attracted attention both synthetically and biologically due to its chemical structure[8] and promising phytotoxic properties[9]. Known to be a non-host specific agonist of the active plant hormone (+)-7-*iso*-jasmonoyl-L-isoleucine (JA-Ile; Fig. 1a)[10], 1 has been found to induce a range of stress-response and defence-related activity in plants by interaction with the jasmonate receptor COR-insensitive 1 (COI1), and inducing phytotoxic effects through activation of the JA-signalling pathway[11]. Through this biological pathway, COR has been reported to exhibit a range of phytotoxic activity across several plant species, including leaf tissue chlorosis[12] and senescence[13], root stunting[14,15], increased ethylene production[16], production of defence-related secondary metabolites[17], induction of hypertrophy[18] and stomatal opening[19]. The jasmonate receptor represents a novel mode of action not currently exploited by commercial phytotoxins, and as such the development of a COR-based herbicide is highly desirable[4].

COR (1) is composed of two constituent natural products: the *cis*-fused 5,6-bicyclic polyketide core unit, coronafacic acid (CFA, 2), coupled to an isoleucine (Ile)-derived amino acid, coronamic acid (CMA, 3), through an amide linkage (Fig. 1a)[8]. Since the discovery of COR 40 years ago[8], considerable synthetic efforts have been directed towards the synthesis of both 2 and 3[20]. However, access to useful quantities of 1 either by bacterial fermentation or synthetically, has been challenging and has afforded only relatively limited structure–activity relationship (SAR) studies (Fig. 1b; vide infra)[21–33]. In addition, 2 has long been viewed as a principal component from which the bioactivity of 1 is derived; however, to date, the reported cumulative production of 2 by chemical synthesis is less than 1 g over nine separate synthesis campaigns. Moreover, hydrolysis of natural 1 is both atom inefficient and prohibitively costly[28,29]. Lastly, there is no substantive quantitative biological data across the intended targets (weed species).

To summarise reported SAR data (Fig. 1b, c), both CFA and CMA moieties confer phytotoxic activity separately, however, this is greatly enhanced when the components are coupled to give the parent structure[24]. With regard to the core moiety, it is known that the *cis*-stereochemistry of the ring junction is important for biological activity, mimicking the side chain configuration of JA-Ile[10,29,34]. Substitution at the C[6] position has also been shown to be required for activity in potato tuber-inducing assays[25]. Reduction of the carbonyl moiety has been reported to lead to reduced volatile inducing activity in rice leaves with respect to COR[27], however; there have been reports of retained activity of this compound. The analogue where the α,β-unsaturated amide has been reduced to afford the fully saturated 6,5-bicycle has been reported and found to be highly active in volatile emission assays[27]. With regard to the amino acid portion, it has been widely reported that the free carboxyl terminus of the amino acid is required for maximal activity[24], and substitution which retains

the S-stereochemistry of CMA at the α-carbon is important for activity, as has been demonstrated through the synthesis of other COR stereoisomers[24]. Tolerance for alternative amino acid substitution with both natural and non-natural amino acids has been demonstrated, however, at the outset of this study, a complete SAR for this portion of the molecule was unclear[29,32].

To enable a comprehensive SAR exploration, a scalable, tractable and flexible synthesis of 2 is required. Herein, we report a collaborative industry-academia approach that has provided a practical, gram-scale synthesis of (±)-2, enabling the subsequent preparation of a >120 member library of analogues of 1. Access to grams of (±)-2 has allowed array synthesis of amide analogues of 1 to explore the binding region around the CMA motif and the flexibility of the synthetic route has allowed SAR charting around the CFA unit, both through single point changes to the bicyclic structure and more significant structural modifications of the core scaffold. This library has been assessed for herbicidal activity against several weed species and, using computational modelling of the active site, has allowed the principal drivers of potency to be revealed, allowing a more rational approach to herbicide discovery using this template.

## Results

**Synthetic strategy**. The synthetic strategy of our collaborative approach[35] was focussed on scalability, to enable preparation of a library of amide analogues of 1 (i.e. variation of the CMA region), and flexibility, to allow SAR interrogation of 1 (i.e. the CFA region). It was our intention that the synthetic campaign and subsequent biological evaluation of COR analogues would inform the design and synthesis of structurally less complex COR derivatives, ideally with the retention or enhancement of phytotoxic potency. Based on the lack of robust SAR data, as the largest fragment, 2 has been assumed to be the key driver of the potency of 1. With the total production of synthetic 2 less than one gram over decades of investigation, access to quantities of this fragment suitable for SAR interrogation has been the most significant challenge in developing COR as an agrochemical lead. As such, our approach had synthetic expediency and flexibility embedded from the outset.

The cyclohexene scaffold of 2 clearly codifies for an intramolecular Diels-Alder (IMDA) disconnection (Fig. 1a) and, indeed, this approach has been used in previous syntheses of (±)-2[36–38]. The requisite diene would be accessed by the aldol disconnection employed by Charette[38]. This ultimately provides a convergent synthesis using two fragments that are readily modifiable, and, therefore, impart the flexibility required of the SAR objectives. The flexibility and mapping strategy of the fragment approach is shown in Fig. 1d. Access to 3 was not problematic and was generated via a modified variant of an established dialkylation process (Fig. 1a)[39].

It has been reported that (+)-1 is significantly more potent than (−)-1 with respect to stomatal opening activity[40]. Accordingly, the SAR associated with each of the stereoisomers was an important aspect of the investigation; however, investment into asymmetric routes at this stage was an inefficient use of resource where the SAR was largely unknown[41]. Based on this, we elected to prepare all compounds as racemates to enable expedient analogue synthesis and, after initial triage in the biological assays, assess single enantiomers by separation of the racemic material.

Our optimised scalable route to (±)-2 is shown in Fig. 2. The aldehyde fragment required for the aldol addition (7) was obtained in five steps and 37% overall yield from the readily available 1,4-butane diol 4[42]. Mono-protection of 4 with dihydropyran proceeded in high yield, allowing isolation of alcohol 5. Swern oxidation afforded the corresponding aldehyde,

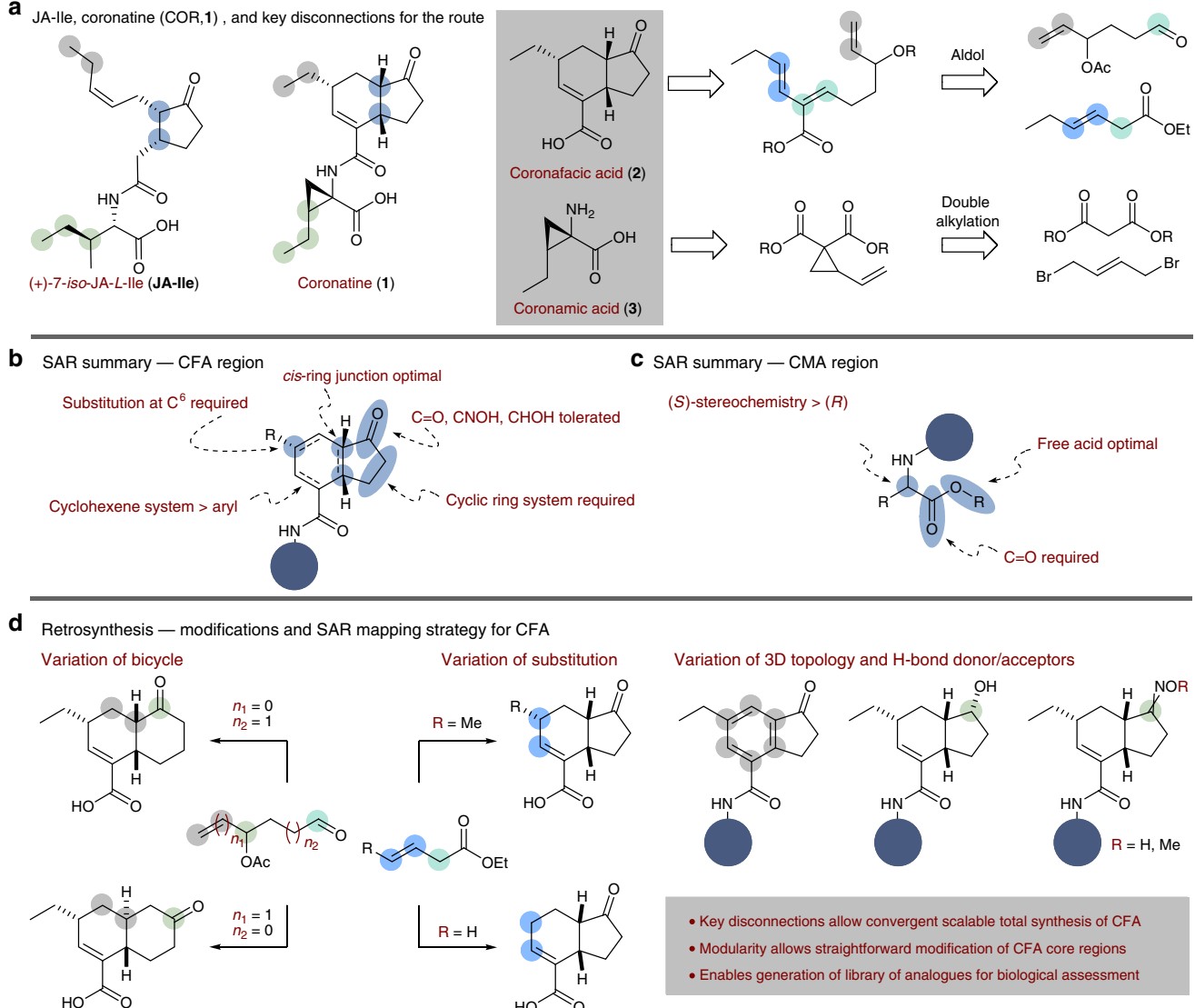

**Fig. 1** Structure, current SAR and route design plan for (+)-7-*iso*-JA-*L*-Ile (JA-Ile) and coronatine (**1**). **a** Structural similarities between the natural bioactive ligand JA-Ile and **1** are highlighted. Coronatine can be considered as comprising of two component parts; the bicyclic core, coronafacic acid **2**, amino acid moiety, and coronamic acid **3**. **b**, **c** A summary of the known SAR at the outset of this work. **d** Retrosynthesis and route design plan for SAR interrogation

which was immediately reacted with vinylmagnesium bromide and quenched with acetic anhydride to give **6** in 63% over 2 steps. THP deprotection followed by further Swern oxidation gave access to aldehyde **7** on multigram scale. The route to this fragment generated in excess of 44 g of **7** for this campaign.

With a robust access to **7**, we then turned our attention to the key aldol addition using ester **8**. Under the cryogenic conditions reported by Charette[38], this reaction predominantly affords the *anti*-product (87:13 *anti*:*syn*). With a view to improving the scalability of this reaction, we observed that allowing the aldol addition to proceed at room temperature gave reversed selectivity, in favour of the *syn*-isomer (83:17 *syn*:*anti*)[43]. This reaction was found to be robust on multigram scale, ultimately allowing access to over 50 g of aldol adduct *syn*-**9**. Latterly, we found *syn*-**9** to be of greater utility than *anti*-**9** in the subsequent dehydration and IMDA reaction.

We had initially viewed the IMDA reaction as being particularly challenging with respect to the scalability of the route. The previously reported requirement of highly elevated temperatures and a pressure-sealed vessel to allow the cyclization

of this class of triene (**10**) is well documented and limits the practicality of carrying out such a procedure on scale[36–38]. However, we found that stereospecific dehydration of *syn*-**9** using CuBr and DIC at moderately elevated temperature afforded the desired *Z*-alkene in situ, which underwent subsequent *exo*-IMDA cyclization in one-pot. While dehydration of *anti*-**9** has been reported[38], this process was less step efficient, requiring isolation of triene **10** prior to the IMDA reaction. Following acyl ester hydrolysis, over 5 g of bicycle **11** was isolated as a mixture of diastereoisomers at C[1], with a *trans*-fused ring junction[36–38]. From **11**, DMP oxidation of the alcohol and acid hydrolysis of the ester, with concurrent epimerization of C[7a], conclude the gram-scale synthesis of (±)-**2** in 10 steps and in 9.9% overall yield. Overall, this route afforded 2.7 g of (±)-**2** to enable the desired analogue synthesis and SAR investigations.

The flexibility offered by this synthetic sequence allowed single point changes to allow the synthesis of a series of CFA analogues (Fig. 1d). Variation of the ester used in the aldol addition (**8**→**14**/**15**), permitted access to analogues bearing structural modification at C[6]. Modification of the cyclopentanone

**Fig. 2** Gram-scale synthesis of (±)-CFA. Five step synthesis of aldehyde **7**, followed by *syn*-selective room temperature aldol addition with ester **8**. Aldol addition product **9** undergoes dehydration and IMDA cyclization of the resultant triene at elevated temperature. DHP: dihydropyran, DMSO: dimethyl sulfoxide, PPTS: pyridinium *p*-toluenesulfonate, DIPEA: *N,N*-diisopropylethylamine, DIC: *N,N'*-diisopropylcarbodiimide, PTSA: *p*-toluenesulfonic acid, PDC: pyridinium dichromate

ring was achieved through use of homologated aldehyde aldol partners (**7**→**12**/**13**), leading to regioisomeric and ring expanded (decalin) cores.

**Library generation**. With access to sufficient quantities of (±)-**2** as well as derivatives with variation of the core CFA template, we prepared a library of analogues to prosecute the SAR objectives (Fig. 3). It has been reported that the enzyme responsible for the linkage of **2** and **3**, coronafacate ligase[44], has a degree of tolerance around the amino acid structure[45], as evidenced through the isolation of several *N*-coronafacoyl compounds alongside COR[46–49]. Accordingly, we determined it appropriate to prepare a range of coronafacoyl amide analogues, maintaining (±)-**2** as the common core unit (Fig. 3a). To ensure breadth in our SAR study, a variety of natural and non-natural amino acids were incorporated using straightforward HATU-mediated coupling on the amino acid methyl esters, followed by hydrolysis under basic conditions to afford the desired carboxylic acid compounds (**16–37**)[32]. For the CFA analogues with single point changes and variation of the carbonyl unit, we prepared both the CMA- and *L*-Ile-derived *N*-coronafacoyl amides using the same amidation procedure (**38–45**), including the decalin and aromatised analogues (**43–45**). Stereoisomers of interest following initial triage (vide infra) were separated by chiral preparative HPLC and evaluated (several examples shown: **46–51**). Lastly, two arrays were generated using automated synthesis with (i) variation of CMA to a range of non-natural amino acids on the aromatised CFA core and (ii) variation of CFA to non-CFA acids on the CMA residue (see Methods and Supplementary Methods).

**Biological assessment**. Overall, 127 analogues of COR were successfully prepared and assessed using a raft of phenotypic assessments against several weed species. A selection of this library is presented in Table 1.

The obtained biological data has allowed mapping of SAR around the natural product scaffold (see Supplementary Tables 7–9 for the full data set):

With regard to the amino acid substituent, there appears to be little tolerance for structural modification away from the CMA motif; variation of this region produced inactive compounds or analogues of significantly reduced potency (e.g., compare **1** vs. **16**, **21**, **32**, **35**, and **36**). This is likely due to the increased bulk in the amino acid region preventing binding to the COI1-JAZ co-receptor[32]. Typically, moderate levels of phytotoxicity were observed with quaternary substituted amino acids, e.g., **32**, aligning this portion of the molecule more closely to the structural features of CMA. In agreement with previous reports, we observed that *S*-stereochemistry at the α-carbon is important for activity, as demonstrated through comparison of **32** and the respective *R*-configured analogue (**31**) which is inactive[24,28]. These results point towards the importance of the amino acid residue to achieve significant levels of potency, and more specifically the importance of a structurally intact CMA unit.

Several of our core-modified CMA-conjugates showed significant phytotoxic activity. As previously mentioned, it is known that the *cis*-stereochemistry of the fused ring junction is important for biological activity[10,29,34]. Compound **43a**, featuring the *cis*-decalin core, showed good levels of activity, however the *trans*-decalin structure **44a** was inactive, implying that the cyclopentenone ring is tolerant of variation but that the *cis*-ring junction is required; however, we cannot rule out that the lack of activity is due to the alternative carbonyl placement (vide infra)[29]. Substitution at the C[6] position is required for activity (**41a** vs. **42a**)[25]. The alcohol derivative, **38a**, showed good levels of activity, while oxime compound **39b** was weakly active, suggesting that variation to the ketone moiety is tolerated; however, this could be a function of metabolism to the ketone in planta. The CMA analogue with aromatic core, **45a**, showed significant levels of activity (Table 1). The activity of this compound also demonstrates the importance of the CMA moiety, as our array analogue

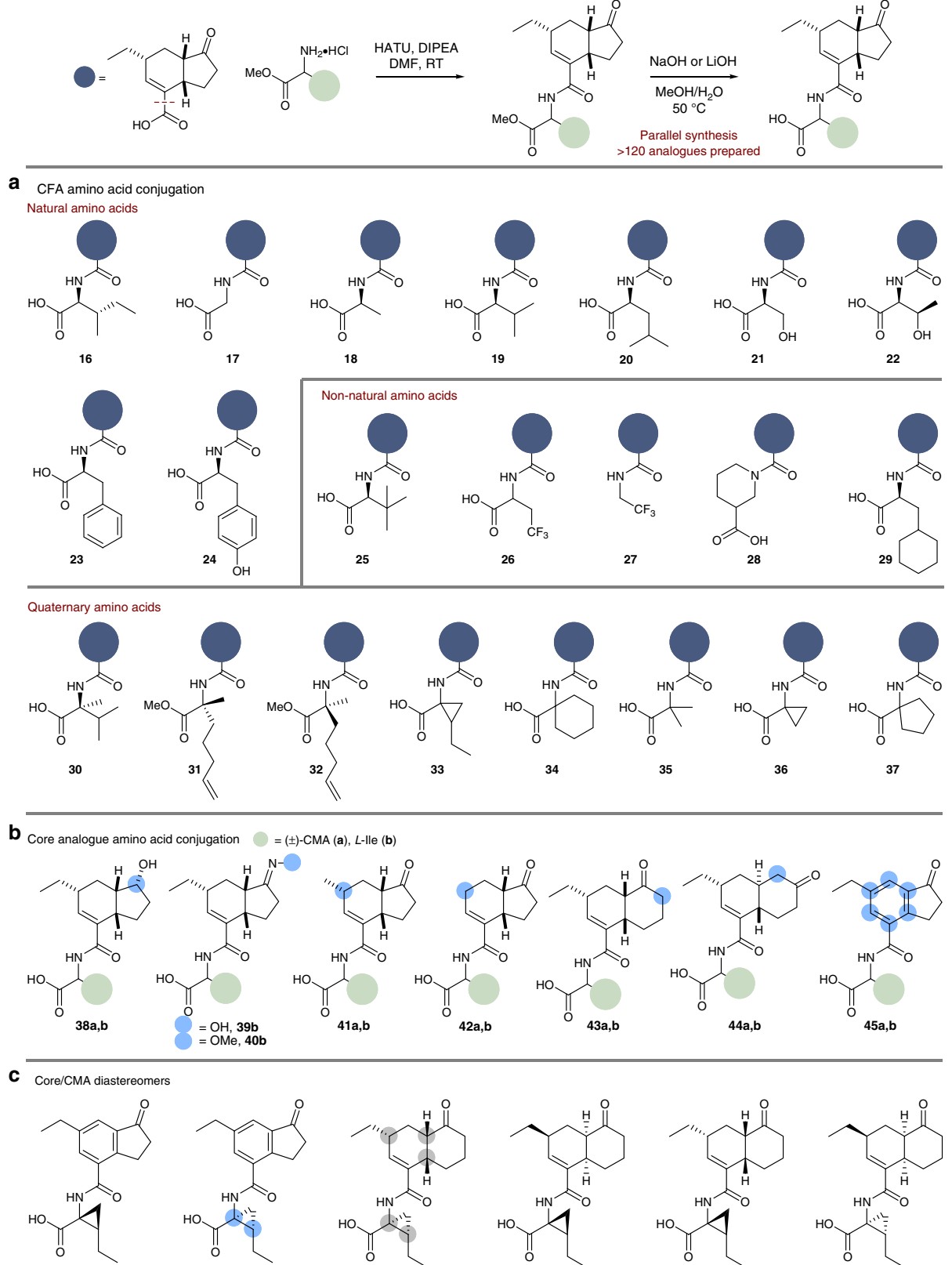

**Fig. 3** Representative examples of COR analogue synthesis. **a** HATU coupling of (±)-**2** and amino acid methyl esters, which were then hydrolysed to afford the free-acids. HATU, 1-[bis(dimethylamino)methylene]-1*H*-1,2,3-triazolo[4,5-b]pyridinium 3-oxid hexafluorophosphate. **b** Alternative core coupling to (±)-CMA and *L*-Ile. **c** CFA core oxime and core analogues

**Table 1 Biological data. Scoring of active compounds from SAR screening**

| Compound | Post-emergence | | | | | Pre-emergence | | | | |
|---|---|---|---|---|---|---|---|---|---|---|
| | AMARE | LOLPE | STEME | DIGSA | Symptom | AMARE | LOLPE | STEME | DIGSA | Symptom |
| (+)-1 | 90 | 60 | NT | 90 | ST/DS | 80 | 90 | NT | 100 | ST/DS |
| (±)-1 | 40 | 0 | 50 | 60 | NC/ST | 70 | 40 | 70 | 80 | NC/ST |
| (±)-2 | 0 | 0 | 0 | 0 | — | 0 | 0 | 0 | 0 | — |
| (±)-3 | 0 | 0 | 0 | 0 | — | 0 | 0 | 0 | 0 | — |
| 16 | 0 | 0 | 0 | 0 | ST | 0 | 0 | 50 | 0 | ST |
| 21 | 0 | 0 | 70 | 20 | NC/ST | 0 | 0 | 0 | 0 | NC/ST |
| 32* | 0 | 0 | 0 | 0 | ST | 50 | 0 | 80 | 0 | ST |
| 35 | 0 | 0 | 0 | 0 | ST | 20 | 0 | 50 | 50 | ST |
| 36* | 50 | 40 | 0 | 60 | ST | 40 | 30 | 50 | 50 | ST |
| 38a* | 70 | 70 | 70 | 80 | NC/ST | 80 | 60 | 80 | 80 | NC/ST |
| 41a | 30 | 20 | 30 | 60 | GI/ST | 0 | 20 | 80 | 0 | GI/ST |
| 43a* | 30 | 10 | 0 | 50 | GI/ST | 30 | 60 | 40 | 80 | GI/ST |
| 45a | 30 | 20 | 10 | 100 | NC/ST | 20 | 20 | 20 | 40 | NC/ST |
| 39b* | 20 | 0 | 40 | 0 | ST | 30 | 40 | 70 | 0 | ST |

In initial greenhouse screening (GH1), compounds are assessed for pre- and post-emergence activity against four weed species, and scored visually for % phytotoxicity (0–100, where 100 is complete control of the target and 0 is no control). Key compounds are designated with asterisks
*AMARE* Amaranthus retroflexus, *LOLPE* Lolium perenne, *STEME* Stellaria media, *DIGSA* Digitaria sanguinalis, *DS* desiccation, *GI* germination inhibition, *NC* necrosis, *NT* not tested, *ST* stunting

synthesis maintaining the aromatic core moiety failed to deliver analogues of significant activity. This result demonstrates the potential for CFA simplification with the retention of phytotoxic activity if the CMA moiety is maintained, a highly desirable outcome of our SAR study as it renders the preparation of 2 unnecessary and replaceable with simpler analogues. The general inactivity of our L-Ile conjugates (38–45b) with CFA analogues in comparison with their CMA substituted counterparts again bolsters importance of the CMA residue. Further attempts to simplify the CFA scaffold with significant structural modifications and with retention of the CMA amino acid were largely unsuccessful.

In light of these results, compounds 43a and 45a were separated into their component enantiomers by chiral HPLC and the single enantiomers were assessed for phytotoxicity. Moderate activity levels were observed; however, in both cases activity levels obtained were weaker than (+)-1. The activity observed from (+)-45a in comparison with the complete inactivity of (−)-45a demonstrates that the potency of 45a is derived from only one enantiomer[40]. This is further demonstrated in the variation of activity levels from the enantiomers of 43a. Of the stereoisomers with the homologated core (48–51), none were more effective than (+)-1, with trends as expected[40].

**Molecular modelling and toxicophore development**. To rationalise these observations, we undertook molecular modelling of COR in the active site of COI1 (Fig. 4). Both the crystal structure[50] and the computational calculations indicate that the binding of COR is chiefly driven by the formation of strong H-bonding interactions with three arginine residues in the active site (ARG85, ARG348, and ARG409) from the amide carbonyl and CMA-based carboxylate group. Favourable interactions also come from the formation of a H-bond with the CFA cyclopentanone carbonyl group and TYR444, as well as a number of hydrophobic interactions, including the ethyl unit of CFA with the lipophilic region consisting of LEU91, PHE89, and ALA86, and insertion of the cyclopropyl-ethane tail into a hydrophobic pocket consisting of ALA384, VAL385, and TYR386.

To rationalise the observed generally detrimental impact of variation of CMA to L-Ile (and the majority of other amino acids), we compared the docking of 1 and 16 (Fig. 4). Docking calculations were performed against the binding site of subunit B

from PDB[51] structure 3OGK[50], using the programme Glide[52,53] in "standard precision" mode. In this approach, the receptor was treated as rigid, while ligands were docked with full conformational flexibility, including sampling of ring conformations, and an additional energy penalty was included to discourage formation of non-planar amide bonds. Following the initial docking, the five highest-scoring poses for each ligand were identified using the Glide "docking score", a metric that accounts for both the ligand-receptor molecular mechanics interaction energy and the ligand strain energy: these five poses were then subjected to a full force-field minimisation, with the resulting, minimised, poses re-scored and only the single highest-scoring docked pose for each ligand retained for analysis. This model indicates that inserting the branched alkyl chain of the Ile unit of 16 into the hydrophobic pocket formed by ALA384, VAL385, and TYR386 is much less favourable than for the CMA unit of 1. The unfavourable binding observed results from multiple steric clashes between the alkyl chain of the Ile unit and the protein. In addition, we expect that the increased flexibility found in Ile vs. CMA would result in a higher entropic cost of binding, negatively impacting on affinity.

With regards to the more tolerant CFA region, the principal interactions are the H-bonding from the CFA cyclopentanone carbonyl group and TYR444 and the hydrophobic interactions from the ethyl unit of CFA with the lipophilic region consisting of LEU91, PHE89, and ALA86. In addition, ARG496 is proximal to the carbonyl of the CFA cyclopentanone; examination of the crystal structure suggests that this is not at a range to form a H-bond; however, we believe the proximity allows this interaction in the dynamic setting. Aromatised compounds (45a) are effective since this placement of functional groups and key interactions are conserved. Removal of either/both of these functional groups induces penalties but so long as they are maintained, there is considerable flexibility in this region, explaining the activity of the decalin and stereoisomeric compounds (38a, 41a, 45a).

Considering all of these data, key potential lead structures that could form the basis of an optimisation programme are 32, 36, 38a, 43a, and 39b (Table 1). The SAR information codified by structures is aligned with the ligand-based toxicophore model described in Fig. 4c, below based on consideration of the biological data obtained. The ligand-based toxicophore model was constructed using Biovia Discovery Studio Visualizer V4.5 (Dassault Systemes, Vélizy-Villacoublay, France). Briefly, the

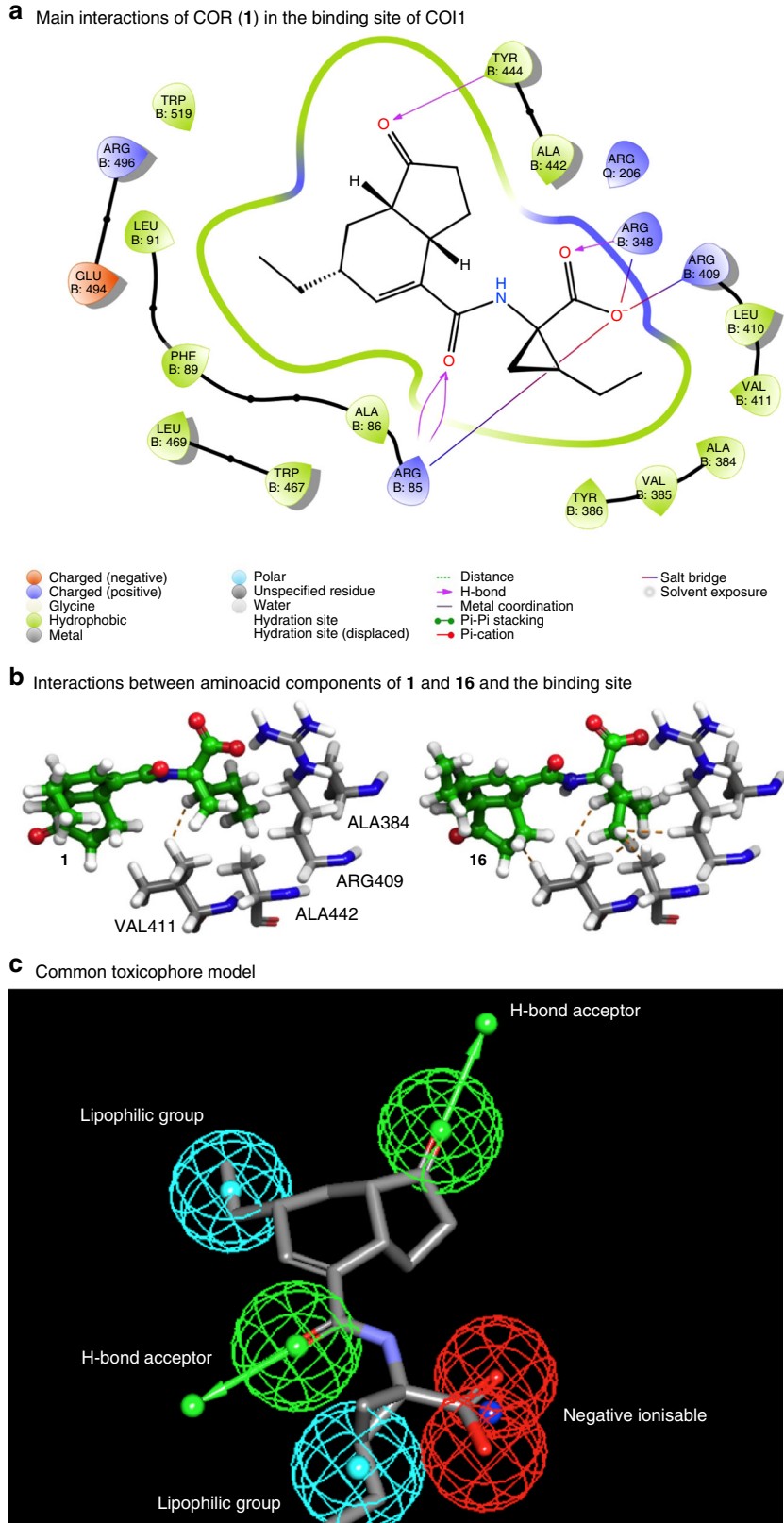

**Fig. 4** Computational modelling. **a** COR (**1**) in the protein binding site (COI1) and main interactions. **b** Docked structures showing the interactions with **1** and **16**. Steric clashes are identified by dashed lines. **c** Common toxicophore model

COR input molecule was energy minimised using a CHARMm force field, and toxicophoric elements were added using relevant point/vector features and constraints based on consideration of the SAR data generated and the available homology modelling information described above.

## Discussion

In conclusion, an extensive SAR investigation around the phytotoxic natural product COR has been carried out. The multigram-scale synthesis of (±)-CFA enabled array synthesis of a broad scoped amino acid screen, using CFA as the common core unit. Investigation of CFA modifications through alterations to the bicyclic structure has allowed for a mapping of SAR around the core-motif. Typically, we have observed that although incorporation of alternative amino acids onto the CFA core can result in low levels of phytotoxic activity, the key convenor of potency appears to be the CMA moiety and a greater tolerance for modification has been observed around the CFA core. These results demonstrate that further studies featuring the CMA unit and CFA replacements may be beneficial, and we would suggest that future research efforts in the area focus on these derivatives.

## Methods

**General methods**. See Supplementary Methods for further details supporting experiments, Supplementary Tables 1–9 for additional data, and Supplementary Figs. 1–339 for spectra.

**Synthesis of 5**. To a round bottom flask was added butane-1,4-diol (**4**) (27.3 g, 302.93 mmol, 5 equiv.) and anhydrous $AlCl_3$ (79 mg, 0.59 mmol, 1 mol%). 3,4-Dihydro-2$H$-pyran (5.42 mL, 59.41 mmol, 1 equiv.) was added slowly and the resulting mixture was warmed to 30 °C for 30 minutes, before being allowed to cool to room temperature. The colourless, crude material was loaded directly in a solution of 40% EtOAc/petroleum ether and purified by flash silica column chromatography, eluent 30−60% EtOAc/petroleum ether to afford **4** as a colourless liquid (9.86 g, 95%).

**Preparation of 6**. To a three-necked flask under an atmosphere of nitrogen was added oxalyl chloride (7.91 mL, 93.48 mmol, 1.5 equiv.) and anhydrous $CH_2Cl_2$ (140 mL). The reaction was cooled to −78 °C and DMSO (13.26 mL, 186.69 mmol, 3 equiv.) added dropwise. The reaction was stirred for 15 minutes at −78 °C before a solution of alcohol **5** (9.81 g, 56.27 mmol, 1 equiv.) in $CH_2Cl_2$ (20 mL) was added dropwise. The reaction was stirred at −78 °C for a further 30 minutes before being quenched slowly with triethylamine (39.6 mL, 284.12 mmol, 5 equiv.). The reaction was allowed to warm to room temperature over 1 h. The pale orange suspension was then diluted with water (40 mL) and extracted with $CH_2Cl_2$ (3 × 30 mL). The organics were combined, washed with brine (20 mL), dried over $Na_2SO_4$, filtered, and evaporated to afford a pale orange liquid. The crude material was loaded directly in a solution of $CH_2Cl_2$ and purified by flash silica column chromatography, eluent 10–20% EtOAc/petroleum ether to afford a pale yellow liquid (7.78 g, 45.00 mmol), which was used immediately in the following step.

Vinylmagnesium bromide (1 M in THF, 45 mL, 45.00 mmol, 1 equiv.) was added dropwise to a stirring solution of the isolated material in anhydrous THF (100 mL) at 0 °C in a three-necked flask under an atmosphere of $N_2$. The resulting solution was allowed to rise to room temperature and stirred for 1.5 h. The reaction was quenched by dropwise addition of acetic anhydride (8.5 mL, 90.09 mmol, 2 equiv.) at room temperature and stirred for a further 1.5 h. The yellow reaction mixture was diluted with water (30 mL) and extracted with EtOAc (3 × 30 mL). The organics were combined, washed with brine (20 mL), dried over $Na_2SO_4$, filtered, and evaporated to afford a pale orange oil. The crude material was purified by flash silica column chromatography, eluent 20% EtOAc/petroleum ether to afford **6** as a colourless liquid (8.65 g, 63%).

**Preparation of 7**. To a round bottom flask was added compound **6** (11.51 g, 47.51 mmol, 1 equiv.) and EtOH (170 mL). PPTS (1.15 g, 4.58 mmol, 0.1 equiv.) was added portionwise and the resulting solution heated to 65 °C and maintained at this temperature for 3 h. The reaction was allowed to cool to room temperature and was then evaporated onto silica gel and purified by flash silica column chromatography, eluent 40% EtOAc/petroleum ether to afford a colourless liquid (5.87 g, 78%).

To a three-necked flask under an atmosphere of nitrogen was added oxalyl chloride (3.32 mL, 39.23 mmol, 1.5 equiv.) and anhydrous $CH_2Cl_2$ (90 mL). The reaction was cooled to −78 °C and DMSO (5.60 mL, 78.84 mmol, 3 equiv.) added dropwise. The reaction was stirred for 15 minutes at −78 °C before a solution of the

alcohol (4.15 g, 26.24 mmol, 1 equiv.) in $CH_2Cl_2$ (10 mL) was added dropwise. The reaction was stirred at −78 °C for a further 30 minutes before being quenched slowly with triethylamine (22 mL, 157.84 mmol, 5 equiv.). The reaction was allowed to warm to room temperature over 1 h. The pale orange suspension was then diluted with water (40 mL) and extracted with $CH_2Cl_2$ (3 × 30 mL). The organics were combined, washed with brine (20 mL), dried over $Na_2SO_4$, filtered, and evaporated to afford a pale orange liquid. The crude material was loaded directly in a solution of $CH_2Cl_2$ and purified by flash silica column chromatography, eluent 10–20% EtOAc/petroleum ether to afford **7** as a pale yellow liquid (3.26 g, 79%).

**Preparation of syn-9**. To a three-necked flask at room temperature under an atmosphere of nitrogen was added ester **8** (2.72 mL, 17.12 mmol, 1.3 equiv.) in anhydrous $CH_2Cl_2$ (50 mL) and DIPEA (3.44 mL, 19.75 mmol, 1.5 equiv.). Dibutylboryltrifluoromethanesulfonate solution (1 M in $CH_2Cl_2$) (17.1 mL, 17.1 mmol, 1.3 equiv.) was added dropwise and the resulting solution stirred at room temperature for 30 minutes. A solution of aldehyde **7** (2.06 g, 13.16 mmol, 1 equiv.) in $CH_2Cl_2$ (10 mL) was then added dropwise and the reaction stirred at room temperature for 1 h. The reaction was quenched with a potassium buffer solution (pH 7.4, 26 mL), MeOH (40 mL) and $H_2O_2$ (30% solution, 13 mL) which were added sequentially. A small exotherm was observed on $H_2O_2$ addition. The reaction was stirred vigorously at room temperature for 16 h, diluted with water (30 mL), and extracted with $CH_2Cl_2$ (3 × 40 mL). The organics were combined, washed with brine (30 mL), dried over $Na_2SO_4$, filtered, and evaporated to afford a pale yellow oil. The crude material loaded directly in a solution of $CH_2Cl_2$ and purified by flash silica column chromatography, eluent 20% EtOAc/petroleum ether to afford syn-**9** as a colourless liquid (2.81 g, 57%).

**Preparation of 11**. To a round bottom flask under an atmosphere of nitrogen was added compound syn-**9** (2.00 g, 6.71 mmol, 1 equiv. (79% purity)), CuBr (96 mg, 0.67 mmol, 10 mol%) and anhydrous toluene (1.3 mL). DIC (1.56 mL, 10.07 mmol, 1.5 equiv.) was added in one portion and the resulting solution was brought to 110 °C for 16 h. The reaction was allowed to cool to room temperature and the crude solution was filtered through celite, eluting with EtOAc (30 mL). The organics were washed with water (30 mL), followed by brine (30 mL), dried over $Na_2SO_4$, filtered, and evaporated to afford a pale brown oil. The crude material was directly loaded in a solution of 10% EtOAc/petroleum ether and purified by flash silica column chromatography, eluent 10% EtOAc/petroleum ether to afford a pale yellow oil (1.49 g, 5.32 mmol), which was not characterised.

To the pale yellow oil was added EtOH (50 mL) and PTSA (mono-hydrate) (1.52 g, 7.99 mmol, 1.5 equiv.) and the resulting solution was brought to 75 °C for 5 h. The reaction was allowed to cool to room temperature and the solvent evaporated to afford an orange oil. The crude material was directly loaded in a solution of 20% EtOAc/petroleum ether and minimal $CH_2Cl_2$ and purified by flash silica column chromatography, eluent 20% EtOAc/petroleum ether to afford **11** as a colourless liquid (677 mg, 54% (2 steps)).

**Preparation of (±)-2**. To a round bottom flask charged with **11** (300 mg, 1.25 mmol, 1 equiv.) in anhydrous $CH_2Cl_2$ (12 mL) was added DMP (794 mg, 1.86 mmol, 1.5 equiv.) in one portion under an atmosphere of nitrogen. The reaction was stirred at room temperature for 16 h before 2 M NaOH (10 mL) was added and the layers stirred vigorously for 10 minutes. The layers were separated and the aqueous further extracted with $CH_2Cl_2$ (2 × 20 ml). The organics were combined, washed with brine (20 mL), dried over $Na_2SO_4$, filtered, and evaporated to afford a colourless oil. The crude material was loaded in a solution of 10% EtOAc/petroleum ether and purified by flash silica column chromatography, eluent 10% EtOAc/petroleum ether to afford a colourless oil (245 mg, 83%).

To a round bottom flask was added compound **S3** (1.10 g, 4.65 mmol) and 3 M HCl (150 mL). The reaction was brought to 100 °C and maintained at this temperature for 16 h. The reaction was allowed to cool to room temperature and extracted with EtOAc (3 × 30 mL). The organics were combined, washed with brine (30 mL), dried over $Na_2SO_4$, filtered, and evaporated to afford an orange oil. The crude material was loaded directly in a solution of 30% EtOAc/petroleum ether and purified by flash silica column chromatography, eluent 30–60% EtOAc/petroleum ether to afford (±)-**2** as a white solid (850 mg, 88%).

**General procedure for synthesis of (±)-1 and analogues**. To a 2-dram vial was added (±)-**2** (30 mg, 0.14 mmol, 1 equiv.) and HATU (66 mg, 0.17 mmol, 1.2 equiv.). DMF (0.7 mL) was added, followed by DIPEA (80 μL, 0.46 mmol, 3 equiv.) and the resulting solution stirred at room temperature for 5 minutes. The amino acid ester (0.21 mmol, 1.5 equiv.) was then added in one portion and the vial capped with a screw top lid. The reaction was stirred for 16 h. The reaction was then diluted with $H_2O$ (10 mL) and the organics extracted with EtOAc (3 × 5 mL). The organics were combined, washed with brine (10 mL), dried over $Na_2SO_4$, filtered, and evaporated to afford the crude product. The crude material was loaded directly in a solution of $CH_2Cl_2$ and purified by flash silica column chromatography to afford the COR ester product.

To a round bottom flask was added the COR ester (24 mg, 0.07 mmol, 1 equiv.) and LiOH (5 mg, 0.20 mmol, 3 equiv.). The material was suspended in 1:1 MeOH: H$_2$O (3 mL) and the resulting suspension brought to 50 °C for 16 h. The reaction was allowed to cool to room temperature, and extracted with EtOAc (1 × 5 mL), and the organics discarded. The aqueous phase was acidified with HCl (aq.), and extracted with EtOAc (3 × 10 mL). The organics were combined, dried over Na$_2$SO$_4$, filtered, and evaporated to afford a colourless oil. The crude material was taken up in a minimal volume of diethyl ether, and petroleum ether added until a white precipitate formed (where precipitation did not occur spontaneously the solvent was concentrated under a stream of compressed air until precipitation occurred). The solvent was removed using a Pasteur pipette and the precipitate dried under vacuum to afford the desired carboxylic acid product.

**Data availability**. All data generated or analysed during this study are included in this published article (and its supplementary information files). These data are also available from the author upon request. Accession codes: The X-ray crystallographic coordinates for structures reported in this study have been deposited at the Cambridge Crystallographic Data Centre (CCDC), under deposition numbers CCDC 1821484 and CCDC 1821485. These data can be obtained free of charge from The Cambridge Crystallographic Data Centre via www.ccdc.cam.ac.uk/data_request/cif. For NMR spectra of the compounds in this article, see Supplementary Figs. 1–339.

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

## Acknowledgements

We thank the EPSRC UK National Mass Spectrometry Facility at Swansea University for analyses, the University of Strathclyde for PhD studentship (M.M.L), and Syngenta for financial and chemical support. We thank Dr M. E. Watson (University of Strathclyde) for supplying amino acids used to prepare **31** and **32**. We thank Thorsten Platz and Russell Ellis (Syngenta) for assistance with compound purification and Matthew Plane and David Pearce (Syngenta) for assistance with robot-enabled parallel synthesis.

## Author contribution

Computational chemistry/docking studies: C.M.B.; Conceptualisation: K.B.L., C.J.R., and A.J.B.W.; Data analysis and analogue design: C.M.B., E.C.F., C.J., K.B.L., M.M.L., M.M.M., C.J.R., and A.J.B.W.; Synthetic chemistry: M.M.L.; Synthetic route design: M.M.L., A.J.B.W.; Project supervision: A.J.D., E.C.F., C.J., C.J.R., and A.J.B.W.; Protein crystal structure refinement: M.G.M.; Toxicophore model: C.J.; Writing of the paper: C.M.B., A.J.D., E.C.F., K.B.L., M.M.L., C.J., and A.J.B.W.; X-ray crystallography: A.R.K.

## Additional information

**Competing interests:** The authors declare no competing interests.

