## [Peer Review File(PDF 329 kb) · Nature Communications]

Reviewer #1 (Remarks to the Author):

In this manuscript, Littleson, et al. describe a large scale racemic synthesis of the phytotoxin coronatine and the synthesis of more than 100 analogs of the parent natural product. While the synthesis uses fairly standard chemistry it is efficient and major hurdles have been solved. The number of analogs generated and the activity trends of these compounds will likely be of interest to those studying the medicinal chemistry of coronatine. The modelling results generally explain the SAR trends observed.

Several points:

The description of the SAR trends is very hard to follow. This is partly because of the large number of compounds synthesized and tested, but also due to poor description of previous work and known SAR trends. For instance, the authors refer to "limited structure-activity relationship studies" in the introduction and cite a dozen papers. Later when discussing SAR trends the authors describe how the results are consistent with previous trends concerning the S-stereochemistry of the alpha carbon. This fact is also surprising given that the authors state that 2 (therefore not the portion of the molecule containing the S-stereochemistry) is the principal component from which bioactivity is derived. It would be helpful for the readers of this manuscript if the previously known SAR trends were summarized more completely (possibly including graphically relative to the structure of 2) in the introduction or elsewhere.

On p. 3 the authors refer to compound 11 when it appears they mean compound 6.

Reviewer #2 (Remarks to the Author):

I am qualified to evaluate only computational part of this work and my comments are only related to this part.

A very good designed work to scale up the synthesis process of Coronatine through gram-scale production of coronafacic acid. It is very satisfactory to see how authors used SAR analysis and reaction pathways for the generation of >120 leads. The work could be published after some minor revisions. I have some comments/suggestions to improve the work.

1. A proper discussion of SAR and docking in the main part of the manuscript will be helpful for readers.
2. It will be interesting to see common pharmacophore approach. If possible, provide pharmacophore structure.
3. The manuscript can be enriched if important functional groups required for enhanced response can be put into a table or figure for better understanding of SAR for beginners and experimentalist.
4. In Figure 4, authors have docked 1 and 4. The docking tool, setting parameters are not discussed. Though it's not typical docking work, but I believe that these basic issues need to be discussed.
5. There must be a list of top 5/10 proposed lead structures considering potency and higher production level.

Comments from Reviewer #1:

The Reviewer requested some additional commentary on the previous SAR. We have added additional text and diagrams as described below:

1. The description of the SAR trends is very hard to follow. This is partly because of the large number of compounds synthesized and tested, but also due to poor description of previous work and known SAR trends. For instance, the authors refer to "limited structure-activity relationship studies" in the introduction and cite a dozen papers.

A very helpful point from the Reviewer. We have added a paragraph describing with a summary of the previous SAR to the top of page 2 and added a diagrammatic summary of this SAR to Figure 1 (Figure 1b). We have adjusted the references and Figure numbering to accommodate these changes.

2. Later when discussing SAR trends the authors describe how the results are consistent with previous trends concerning the S-stereochemistry of the alpha carbon. This fact is also surprising given that the authors state that 2

(therefore not the portion of the molecule containing the S-stereochemistry) is the principal component from which bioactivity is derived.

*There may be some confusion here; this may have arisen due to the limited discussion of previous SAR (see above). If we understand the Reviewer's comments correctly, the Reviewer is referring to the following statement from below Table 1: "In agreement with previous reports, we observed that S-stereochemistry at the α -carbon is important for activity, as demonstrated through comparison of **32** and the respective R-configured analogue (**31**) which is inactive (See ESI)." This statement refers to the amino acid portion, i.e., CMA (**2**) (see the preceding text for context). The parent structure (CMA) contains (S)-stereochemistry at the alpha-carbon and it is this stereochemistry that is required for activity. To ensure this is clearer, we have added additional description of the previous SAR (top of page 2).*

3. It would be helpful for the readers of this manuscript if the previously known SAR trends were summarized more completely (possibly including graphically relative to the structure of **2**) in the introduction or elsewhere.

As described above (1), we have added a diagrammatic representation of the previous SAR to Figure 1 (as Figure 1b).

4. On p. 3 the authors refer to compound **11** when it appears they mean compound **6**.

My apologies for this error. This has been corrected.

Comments from Reviewer #2:

The Reviewer requested some additional commentary on the previous SAR and the SAR generated in our study. We have added additional text as described below:

1. A proper discussion of SAR and docking in the main part of the manuscript will be helpful for readers.

As described for Reviewer 1, we have added a paragraph describing with a summary of the previous SAR to the top of page 2 and added a diagrammatic summary of this SAR to Figure 1 (Figure 1b). We have adjusted the references and Figure numbering to accommodate these changes.

2. It will be interesting to see common pharmacophore approach. If possible, provide pharmacophore structure.

This is a very helpful point by the Reviewer. We have developed a toxicophore model of the ligand and added this to Figure 4, as well as providing the methods used to establish this model (page 9).

3. The manuscript can be enriched if important functional groups required for enhanced response can be put into a table or figure for better understanding of SAR for beginners and experimentalist.

As described above, we have added a diagrammatic summary of this SAR to Figure 1 (Figure 1b).

4. In Figure 4, authors have docked **1** and **4**. The docking tool, setting parameters are not discussed. Though it's not typical docking work, but I believe that these basic issues need to be discussed.

We have added a full description of this docking work to page 8.

5. There must be a list of top 5/10 proposed lead structures considering potency and higher production level.

Another very helpful point. We have highlighted the top 5 compounds in Table 1, with amendments to the legend as appropriate, as well as discussing these compounds in their relation to the common toxicophore model/SAR on page 9.

Reviewer #1 (Remarks to the Author):

This is a solid paper that demonstrates the ability of chemical synthesis to produce a) a substantial quantity of a biologically important secondary metabolite and b) provide large numbers of analogs of the metabolite for SAR evaluation. The revisions made by the authors make the paper read more clearly and should provide the reader with a better idea of what was previously known. The revisions would seem to address the concerns of the reviewers.

The following minor change should be made before publication:

The dashed bonds are pointing the wrong way on compounds 31 and 32.

Reviewer #2 (Remarks to the Author):

The authors addressed all my concerns.

I do recommend the revised manuscript for publication.

Dear Dr. Bottari,

Many thanks for your email. We are delighted that our manuscript has been accepted for publication in *Nature Communications*.

As requested, a full list of the comments and requests from the Reviewers is provided below along with our response and description of any changes made.

We have uploaded two versions of the revised manuscript file. One of these has 'Track Changes' so that you can more readily see and verify the changes we have made.

Please let me know if I can provide any further information or clarification.

I'd like to take this opportunity to thank you and the Reviewers for your assistance with this submission.

Best wishes,

Allan

REVIEWERS' COMMENTS:

Reviewer #1 (Remarks to the Author):

This is a solid paper that demonstrates the ability of chemical synthesis to produce a) a substantial quantity of a biologically important secondary metabolite and b) provide large numbers of analogs of the metabolite for SAR evaluation. The revisions made by the authors make the paper read more clearly and should provide the reader with a better idea of what was previously known. The revisions would seem to address the concerns of the reviewers.

The following minor change should be made before publication: The dashed bonds are pointing the wrong way on compounds **31** and **32**.

Response: We thank the Reviewer for bringing this to our attention. This has been addressed in the submitted files.

Reviewer #2 (Remarks to the Author):

The authors addressed all my concerns. I do recommend the revised manuscript for publication.

Response: We thank the Reviewer for their support of our manuscript.